# Natural Fiber-Reinforced Composite Incorporated with Anhydride-Cured Epoxidized Linseed-Oil Resin and Atmospheric Pressure Plasma-Treated Flax Fibers

**DOI:** 10.3390/ma17174244

**Published:** 2024-08-28

**Authors:** Sofya Savicheva, Bastian E. Rapp, Nico Teuscher

**Affiliations:** 1Laboratory of Process Technology, NeptunLab, Department of Microsystems Engineering (IMTEK), University of Freiburg, Georges-Köhler-Allee 103, 79110 Freiburg im Breisgau, Germany; bastian.rapp@neptunlab.org; 2Fraunhofer Institute for Microstructure of Materials and Systems IMWS, 06120 Halle (Saale), Germany; nico.teuscher@imws.fraunhofer.de

**Keywords:** flax fibers, natural fiber-reinforced composite, epoxidized linseed oil, atmospheric pressure plasma

## Abstract

Fiber-reinforced composites (FRCs) represent a promising class of engineering materials due to their mechanical performance. However, the vast majority of FRCs are currently manufactured using carbon and glass fibers, which raises concerns because of the difficulties in recycling and the reliance on finite fossil resources. On the other hand, the use of natural fibers is still hampered due to the problems such as, e.g., differences in polarity between the reinforcement and the polymer matrix components, leading to a significant decrease in composite durability. In this work, we present a natural fiber-reinforced composite (NFRC), incorporating plasma pre-treated flax fibers as the reinforcing element, thermoplastic polylactic acid (PLA) as a matrix, and a key point of the current study—a thermoset coating based on epoxidized linseed oil for adhesion improvement. Using atmospheric plasma-jet treatment allows for increasing the fiber’s surface energy from 20 to 40 mN/m. Furthermore, a thermoset coating layer based on epoxidized linseed oil, in conjunction with dodecyl succinic anhydride (DDSA) as a curing agent and 2,4,6-tris(dimethyl amino methyl) phenol (DMP-30) as a catalyst, has been developed. This coated layer exhibits a decomposition temperature of 350 °C, and there is a substantial increase in the dispersive surface-energy part of the coated flax fibers from 8 to 30 mN/m. The obtained natural fiber-reinforced composite (NFRC) was prepared by belt-pressing with a PLA film, and its mechanical properties were evaluated by tensile testing. The results showed an elastic modulus up to 18.3 GPa, which is relevant in terms of mechanical properties and opens up a new pathway to use natural-based fiber-reinforced bio-based materials as a convenient approach to greener FRCs.

## 1. Introduction

Composite materials are a highly versatile type of engineering material that results from “combining two or several components, including reinforcing fibers and a polymer matrix, to achieve specific properties” [1,2]. The matrix determines the composite’s shape, the appearance of the surface, resistance to environmental processes, and general durability, whereas the embedded fibers are responsible for bearing the structural loads and providing stiffness and strength [3]. Traditionally, carbon fibers have been the default choice for this purpose in lightweight construction. However, due to their reliance on petroleum resources, non-renewability, and high energy consumption, there is a growing need for more sustainable alternatives [1]. Natural fibers have become a viable option due to their renewability and various options for pre- and post-treatment [2]. Their lighter weight brings advantages like reduced fuel consumption and improved recycling opportunities. Natural fiber-reinforced composites (NFRCs) are commonly used in the automotive industry, particularly in door panels and linings, often combined with materials like epoxy, thermoset resin, PLA, and polypropylene (PP). Beyond the automotive sector, fiber composites have the potential for broad applications, ranging from aircraft components and electronics to construction [4]. The forecast for the consumption of natural fiber composites worldwide, prepared by Beijing Yubo Zhiye Market Consulting Co., showed three significant regions for the development of natural fiber composites: North America, China and Europe, from approximately 200 ten thousand ton in 2021 to near 300 in 2026 [5]. A crucial factor that significantly influences the final performance of NFRCs is the interface between the fiber and matrix [2]. Natural fibers are composed of cellulose and hemicellulose, which contain hydroxyl groups that contribute to the fibers’ hydrophilic and polar characteristics, while most common matrix polymers are significantly more hydrophobic [6]. A composite based on natural fibers bears significant risks of moisture uptake due to the high polarity of the hydrophilic cellulose [7,8]. This weakens the bond between the fiber and matrix, causing a reduction in interfacial adhesion and thus leading to microcracks in the matrix and further composite degradation [7]. Another factor that could influence the final composite performance is the reinforcement architecture of the fibers themselves [9] or their lignin content [8]. Additionally, wax and pectin, both of which are ample in the plant fiber’s surface, hinder interlocking with the polymer matrices, further degrading interfacial performance [7]. Therefore, it is essential to modify the fiber surface chemically, e.g., by alkali treatment [10], silane treatment [11], acetylation [12] or via corona or plasma treatment, to adjust the surface energies between the fibers and the matrix. Among all techniques, plasma treatment, using either inert or reactive gases, provides a mild and physical treatment option for the activation of the fiber surface, thus further improving the sustainability footprint of the material [13].

In addition to the above-mentioned chemical modification of the fiber surfaces, the application of a functional interface layer between the fiber and the matrix may enhance the bonding between them, thus avoiding microcracks in the matrix and further composite degradation. Although thermoset matrices based, e.g., on epoxy chemistry, offer several advantages, including, e.g., exceptional thermal and mechanical resilience and, notably, excellent water resistance [3], these resins primarily rely on bisphenol A, a molecule subject to considerable ecological and human health concerns [14]. Therefore, employing greener alternatives is a viable option to improve the carbon footprint and the sustainability of these composites. Consequently, in recent years, we have seen a significant increase in the interest in developing resins derived from renewable sources, such as, e.g., vegetable oils [14]. Resins based on epoxidized plant oils not only meet environmental standards but also offer superior mechanical properties, comparable processing conditions, and reduced material brittleness compared to bisphenol A-based alternatives [15]. In addition to the matrix itself, the curing system of epoxides is worth investigating. Most of the epoxy thermoset systems are prepared by hetero-polymerization mechanisms, utilizing alkaline curing agents such as primary or secondary aliphatic amines or incorporating hardeners such as, e.g., anhydrides [16]. Despite their advantages such as, e.g., a long pot life or low viscosity, most of these are solids and therefore require elevated temperatures for the resin preparation [17,18]. To solve this challenge, liquid anhydrides, such as nadic methyl anhydride (NMA) or dodecyl succinic anhydride (DDSA), could be used due to their miscibility with epoxy resins at room temperature [18]. In general, these systems usually require a nucleophilic accelerator [18].

In this work, we describe a NFRC based on flax as the fiber material, using polylactide (PLA) as the matrix and epoxidized linseed oil (ELO) as the interface layer between the flax and PLA. This thermoplastic toughening of ELO-based resins (which are used primarily for the interface layer but not as the bulk) has the potential, in a way, to improve crack-propagation resistance [19] and enable us to prepare an ultra-lightweight reinforced material [20,21]. In order to adjust the surface energy of flax fibers, a mild treatment based on atmospheric plasma was investigated. For the interface layer, we used DDSA as the curing reagent, which is convenient to handle due to its liquid state and long aliphatic chains, which generally reduce the brittleness of the final composite [18]. As the curing agent of the ELO-based resins, we used tris(dimethylaminomethyl) phenol (DMP-30), a tertiary amine. The three-step processing is shown in Figure 1a. Firstly, the flax fibers are modified using an atmospheric-pressure plasma treatment, during which the surface free energy (SFE) of the raw flax was increased from 24 to 35 mN/m. Secondly, a thermoset coating based on epoxidized linseed oil was created as a thin layer, with a dispersive surface energy up to 31 mN/m, which serves as an interface layer between the fibers and the matrix. Lastly, the coated fibers were embedded in PLA as the main matrix material.

## 2. Materials and Methods

### 2.1. Materials

Raw flax tape (density 110 g/m^3^, unidirectional) was supplied by Ecotechnilin (Valliquerville, France), and non-woven flax (density 270 g/m^3^, cross-stitched) was supplied by Ecco Gleitwing (Tokyo, Japan). Epoxidized linseed oil (oxirane content 8.5–9.5%) [22] as a resin was purchased from Hobum Oleochemicals GmbH (Hamburg, Germany). Moreover, 2-Dodecenyl succinic-acid anhydride (90% purity) as a curing agent and 2,4,6-Tris(dimethylaminomethyl) phenol (95% purity) as catalyst were purchased from SERVA (Heidelberg, Germany). Luminity L105 polylactic acid neat resin (density 124 g/m^3^, melting temperature 175 °C) was purchased from Total Corbion (Gorinchem, The Netherlands). All products were used as received.

### 2.2. Methods

#### 2.2.1. Plasma Pretreatment

In this work, two types of flax were investigated: raw flax tape, as the material of interest (named shorty as M_1_), and non-woven flax as a reference sample (named M_2_), because of its flat fiber-bonded surface free from organic contaminants [23]. The fiber surface was treated by an atmospheric pressure plasma jet (APPJ) using the Arcojet PG 051P (Arcotec, Mönsheim, Germany). We investigated two parameters during the plasma treatment, i.e., the number of treatment cycles (one cycle is a full sweep of the sample at a speed of 2.6 m/min) and the type of gas used during plasma treatment. Here, air plasma is used for the reactive etching of the surface, i.e., for increasing the surface roughness and introducing oxygen surface functional groups. As an alternative, nitrogen was investigated, which allows for introducing nitrogen functional groups such as, e.g., amines [10]. The flowrate for air and nitrogen were 90 and 80 ltr/h, respectively.

#### 2.2.2. Resin Preparation

ELO was diluted with DDSA and DMP-30. The resulting mixture was homogenized at an ambient temperature in an ultrasonic stirrer for 5 min. The ratio of the components—ELO:DDSA:DMP-30—was fixed at 100:45.45:6 in fractions by weight. The curing speed of this system at room temperature is negligible and requires elevated processing temperatures [17].

#### 2.2.3. Coating Application

The flax substrates were placed inside a chamber and spray-coated with a spray gun (decor spray-gun W3/W3F by Alfred Schütze Apparatebau GmbH, Weyhe, Germany) using nitrogen at a pressure of 1 bar. The distance between the sample and the nozzle was fixed at 10 cm, and spraying was performed for 20 s. The coated fiber surface was subsequently cured at 180 °C for 20, 30 and 40 min in an oven (UT12 by Thermo Heraeus, Hanau, Germany).

#### 2.2.4. Preparing the NFRC

For the tensile test, the laminates with a thickness of 1 mm were prepared. The fiber layer was stacked between PLA films in a 0° direction and laminated at 200 °C, a pressure of 1 bar, and at speeds of 1 and 2.5 m/min by a double-band press Meyer KFK-E 1100 (Herbert Meyer Maschinenfabrik GmbH, Roetz, Germany) for continuous laminate production. The fiber content was held constant at 40 wt. %.

#### 2.2.5. Characterization

Contact angle (CA) measurements were performed using a DSA100M drop shape analyzer (Krüss, Hamburg, Germany). The calculation of the surface free energy of samples and the polarity contribution were carried out using Advance 1.17-01 software by Krüss. For the evaluation of surface tension, two types of liquids with known polar and dispersed parts were used. In the case of raw flax tape, water and white oil were used, while for the non-woven flax, ethylene glycol was chosen as a polar liquid. Since the surface of the reference material was pretreated by the manufacturer, it was free from fats or other organics that usually prevent fast water absorption. The reported values of the CA are the mean value and standard deviations of ten measurements at ten different locations of the samples, respectively. The characterization of the surface chemistry of plasma-treated fiber surfaces was conducted by utilizing Attenuated Total Reflection–Fourier Transform Infrared (FTIR-ATR) using a Routine Spectrometer (Bruker, Ettlingen, Germany) on spectra from 4000 to 400 cm^−1^ (recording spectra: 64, resolution: 4 cm^−1^) and analyzed using OPUS 8.5 (SP1) Build 8.7.10 software (Bruker, Germany). Thermogravimetric analysis (TGA) and differential scanning calorimetry (DSC) measurements were carried out on an STA 449 F1 Jupiter (Netzsch, Waldkraiburg, Germany), using Proteus 5.2.1 software (Netzsch, Germany). To characterize the thermal stability, the sample were placed in alumina vessels and heated at 10 K/min from 0 to 600 °C with a helium flow rate of 30 mL/min. Indium standards were used for the calibration of temperature and enthalpy. Static tensile tests were conducted at 23 °C using a 2.5 kN load cell on a Zwick Z050—ZwickRoell (Ulm, Germany). Following the DIN EN ISO 527 standard [24], the tensile strength of the prepared composites was assessed at a testing speed of 2 mm/min. Each specimen measured 150 mm in length and 15 mm in width, with an external clamp gauge (makroXtens—ZwickRoell) length of 50 mm and a length of 110 mm between the grips during testing. For each set of conditions (curing times and processing speed of the press), three samples were prepared. The morphology of the cryofractured surfaces of the materials was examined by scanning electron microscopy (SEM) using a FEI Quanta 3D FEG (GFZ Helmoltz centre Potsdam, Potsdam, Germany), with an Everhart–Thornley detector and an acceleration voltage of 5 kV.

## 3. Results and Discussion

### 3.1. Plasma Pretreatment of the Fibers

The main motivation for the plasma pre-treatment was to achieve surface cleaning (to remove organic contaminations such as waxes or fats) and surface functionalization to improve interface strength, increase the surface area, and activate all bonding mechanisms responsible for adhesion—mechanical, physical, and chemical [25]. As stated, different surface functionalization types were expected from atmospheric and nitrogen gas. Increasing the surface free energy (SFE) through the introduction of polar surface groups improves the wetting of the fibers by the matrix material [26,27]. Figure 2 shows that the SFE was increased for both flax types under all treatment conditions, reaching increases from approximately 33 to 40 mN/m for non-woven flax and from 24 to 35 mN/m for raw flax tape (Appendix A). Based on the experimental results, which show relatively low SFE values for the raw flax after several treatments and nearly equivalent SFE results with non-woven flax after 10 cycles, we assume that the outer hydrophobic layer of the tested fiber surface is likely removed prior to the attachment of functional groups [28,29].

The SFE energy increase is due to the introduction of chemical functional groups, such as ketone, carboxylic, aldehyde, and nitro groups on the surface [27], which we investigated using FTIR-ATR spectroscopy (Figure 3). The two main wavenumber ranges where the chemical change was noticed were 2850 and 1600 cm^−1^. These bands are attributed to the cleaning of the surface by plasma treatment [29] and thus correlated to the removal of methyl and methylene groups of waxes (peaks 2860 and 2846 cm^−1^) and the decrease in aromatic groups of lignin or change in water content, respectively. According to the FTIR analysis, utilizing air plasma offers a stronger effect compared to pure nitrogen, which is most likely due to the longer radiative lifetime, according to the data on the required excitation energy and radiative lifetime of particles [13,29].

### 3.2. Characterization of ELO Resin

Differential scanning calorimetry (DSC) was carried out in order to establish the curing temperatures of the epoxy resin. Complete curing was found to be achieved at 250 °C after exothermal onset starting at 108 °C. The decomposition of the resin was found via thermal gravimetric analysis (TGA) at around 358 °C, with an overall mass loss of 98.9%. The summarized data of DSC and TGA tests are presented Figure 4 and Table 1.

In order to further elucidate the adhesion compatibility of the components in the composite, we compared SFE values of pure (plasma untreated), treated fibers and ELO-coated fibers (Figure 5). The high polar part of the SFE of plasma-treated fibers (36.5 mN/m polar part and 7.47 N/m dispersive part) leads to better adhesion strength [26] and improves compatibility with the ELO coating in the next step of composite preparation. The SFE of ELO-coated fibers showed rather hydrophobic behavior, an 31.49 mN/m dispersive part and an 0.01 N/m polar part, while SFE values of the fibers, which went through all preparation steps, such as plasma treatment, ELO coating and then plasma treatment again, became similar to those of the not-treated one (SFE of 28.78 mN/m compared with SFE of not treated 32.86 mN/m). Since one of the aims of this work was to achieve good adhesion between hydrophilic fibers and the relatively hydrophobic thermoplastic PLA matrix (water CA ~70°) [30], further plasma treatment after ELO coating was not necessary to proceed.

### 3.3. Mechanical Characterization of the Prepared NFRC

The mechanical properties of the prepared composites are summarized in Table 2. For the mechanical evaluation, the final NFRC samples were measured with the following parameters to find the optimal composite preparational conditions:Different precured times (20, 30 or 40 min) for the ELO coating.Different proceeding speeds of the belt pressing for PLA infusion (1 m/min—referred to as I; 2.5 m/min—referred to as II).

As can be seen from Figure 6, the composite containing the ELO coating cured for 20 min, which was further belt-pressed with PLA at speeds of 2.5 m/min, showed the highest elastic modulus value, i.e., 18.3 GPa (Appendix A).

For a better understanding of the mechanical performance of the composite, the tensile test results were compared with values reported in the literature. The Young’s modulus of the prepared ELO composite was found to be higher than the 7.3 GPa observed for PLA/40 wt. % flax composites [31]. The same authors studied the effect of adding a triacetin plasticizer (Tri) to PLA, which did not enhance mechanical properties (7.3 GPa for PLA/5 wt. % Tri/40 wt. % flax). Both of these composites were prepared using extrusion—a process that can subject fibers to shearing, potentially leading to fiber degradation and lower mechanical performance [32]. In the current study, the fibers were subjected to multiple heating cycles during ELO curing and belt-pressing, yet the results remained relatively high and should be carefully considered in each specific case. The high standard deviation (S.D.) of tensile stress could be explained by not-uniform loading during the composite preparation and could be improved by varying manufacturing techniques, compression modeling, or using a hand lay-up ore belt-press.
materials-17-04244-t002_Table 2Table 2The mechanical properties of prepared NFRC.Material(Curing Time/Speed)Young’s Modulus [GPa]S.D.Tensile Stress [MPa]S.D.Elongation to Break [%]S.D.Ref/I18.30.3195.52.11.20.1Ref/II18.12.7182.27.41.20.120 min/I14.81.8162.541.10.120 min/II18.30.2171.14.610.0430 min/I15.91.6147.712.40.80.230 min/II17.70.4147.41.20.90.0140 min/I15.92.7131.74.50.90.240 min/II16.50.6143.81.910.04

The tensile properties of NFRCs are mostly affected by the interfacial adhesion between the resin and fibers and the fiber volume fraction in the matrix resin. The value of the latter could be determined by a fiber content of 40–50 wt. % for an injection-molded thermoplastic matrix [33]. Also, this leads to the load distribution moving to the fibers and not the matrix; therefore, there is a decreased risk of failure of the whole composite [6].

To obtain detailed topographical information about the adhesion interphase, cryofractures of the composite were generated and analyzed by SEM (Figure 7). As can be seen, fibers had a tendency to aggregate in bundles; this could be a consequence of not-uniform ELO coating and became one of reasons for the decreasing mechanical performance of the composite. Additionally, failure at the fracture is due to bulk material breaking, as no fiber/matrix debonding was observed.

## 4. Conclusions

In this work, we demonstrated the suitability of atmospheric plasma-treated natural fibers as a suitable material for generating all-bio-based NFRCs. The effect of the gas type and time of treatment on surface functionalization was evaluated. The results showed an SFE increase from 20 to 40 mN/m and also a change in the chemical group content on the fiber surface—that is, the removal of organic contaminations, which is used to prevent interlocking. Secondly, we introduced an interface layer based on a green alternative epoxy system, i.e., ELO, which enhances surface adhesion between the hydrophilic fibers and a relatively hydrophobic thermoplastic matrix material, PLA. For the preparation of the ELO resin, we used the tertiary amine DMP-30 as a catalyst and DDSA as a curing agent. As the latter is a liquid miscible in epoxy resin anhydride, the formulation of the resin is a room-temperature process, thus enhancing the economic and environmental benefits of the system. ELO coatings of the fibers afforded a highly dispersive SFE, significantly improving wetting and adhesion by and with the hydrophobic thermoplastic PLA film. The SEM analysis of cryofractured NFRCs after belt pressing showed good adhesion between the fibers and matrix.

This work might be instrumental for the further development in the field of NFRCs, which have thus far become a viable alternative to conventional carbon fiber-based FRCs, enhancing the sustainability and economic feasibility of this important high-performance material system.

Future challenges in the development of NFRCs could focus on enhancing thermoset preparation, such as reducing the amount of anhydride and catalyst, and improving the reactivity of the main thermoset component (epoxidized linseed oil) by replacing it with more chemically reactive acrylated epoxidized plant oil, which contains not only epoxy rings but also hydroxyl groups [34,35,36,37]. Additionally, conducting fracture tests on anti-parallel oriented laminates could provide deeper insights into their mechanical performance [38].

## Figures and Tables

**Figure 1 materials-17-04244-f001:**
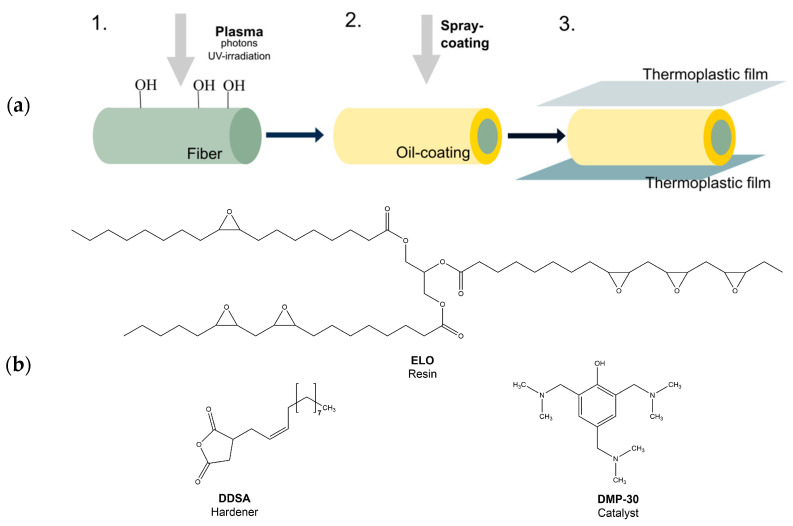
(**a**) Schematic of the preparation of natural fiber-reinforced composite (NFRC): 1. Plasma pretreatment of the fiber surface; 2. Application of the fiber coating based on epoxidized linseed oil; 3. Infusion of the fibers with PLA as matrix material; (**b**) Monomers for the resin formulation: ELO, DDSA and DMP-30.

**Figure 2 materials-17-04244-f002:**
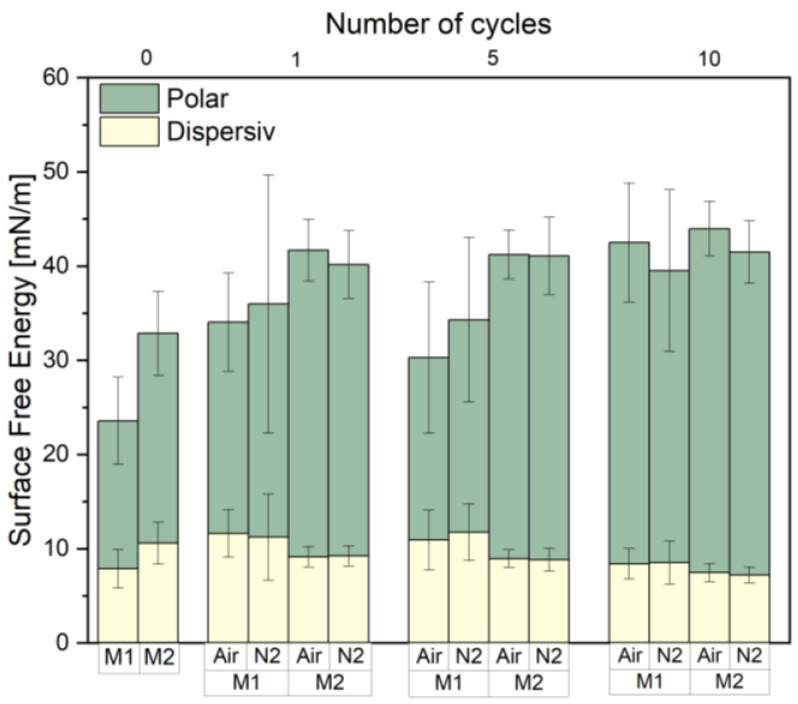
Influence on the increase in SFE as a function of cycle number and gas type during plasma treatment of both flax types. Higher increases in SFE are found with higher cycle numbers. Air plasma yielded higher SFE increases in all cases. Displayed data are mean and average of 10 measurements at different locations on the sample.

**Figure 3 materials-17-04244-f003:**
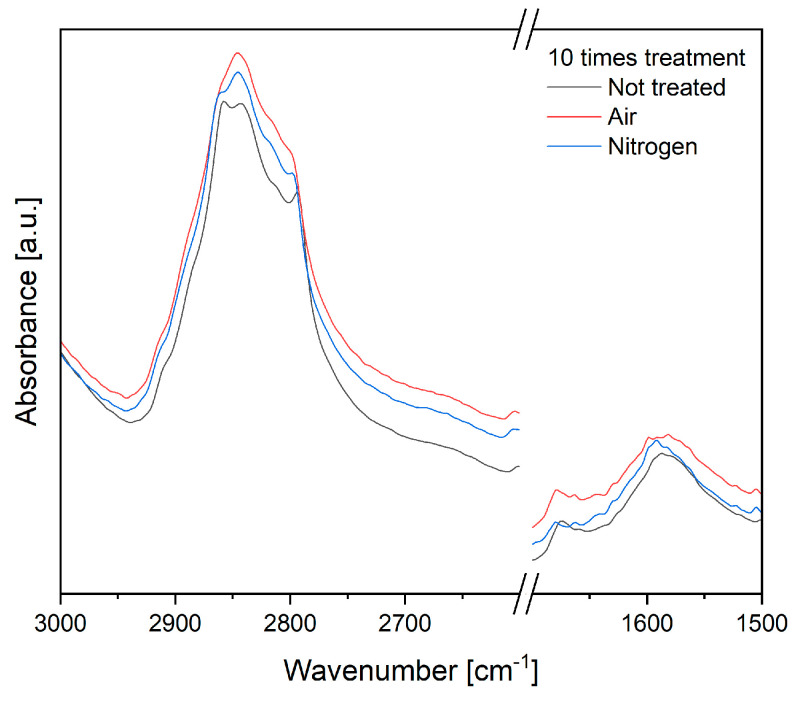
Characterization of the surface chemical structure of non-woven flax by FTIR-ATR. Comparison of spectra of samples treated for 10 cycles by air or nitrogen plasma. Stronger effects are found for the reactively treated samples, i.e., using air plasma.

**Figure 4 materials-17-04244-f004:**
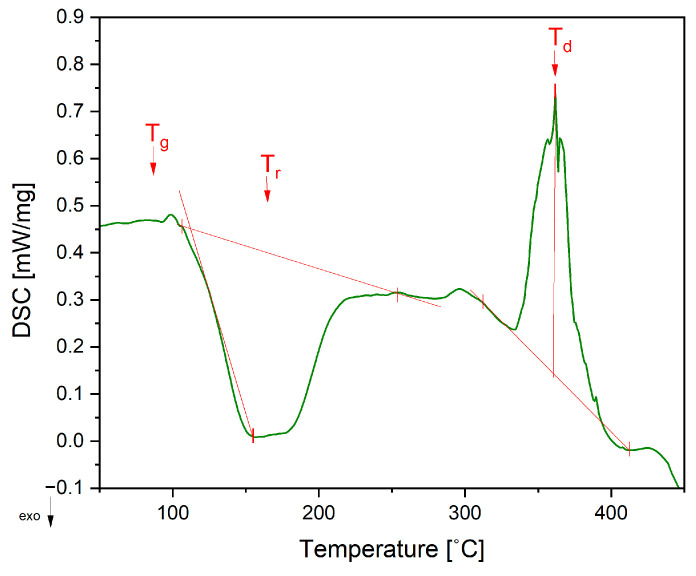
DSC thermogram of the ELO resin displaying glass transition (T_g_), recrystallization (T_r_) and decomposition (T_d_).

**Figure 5 materials-17-04244-f005:**
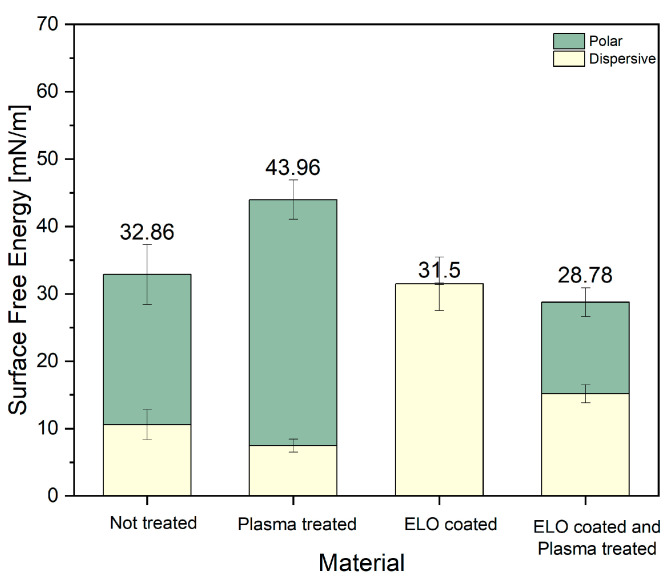
SFE comparison of non-treated, plasma-treated sample, ELO-coated fibers before and after plasma treatment.

**Figure 6 materials-17-04244-f006:**
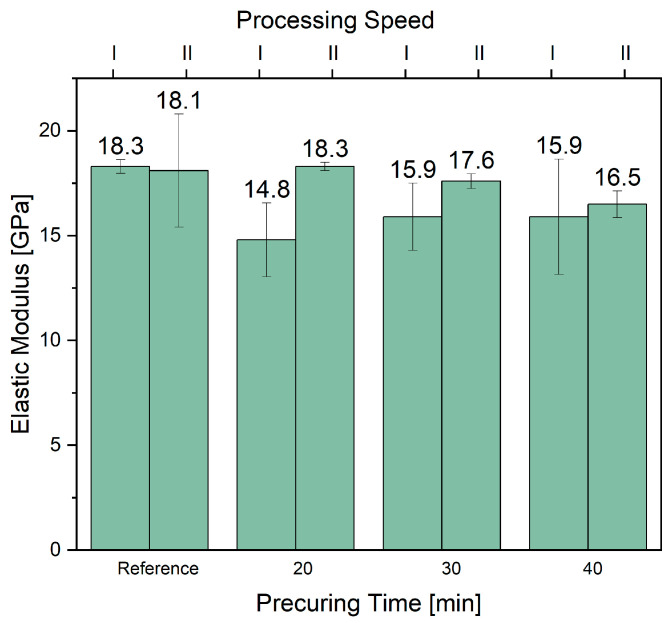
Young’s modulus of the prepared NFRC prepared varying the curing times of the ELO coatings as well as speed of the belt process (1 m/min—I; 2.5 m/min—II).

**Figure 7 materials-17-04244-f007:**
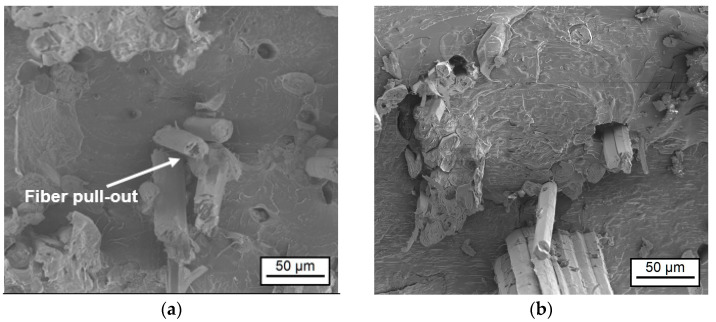
SEM micrographs of cryofractured NFRC: (**a**) Reference sample with plasma-treated fibers (five cycles under the air). Further figures: plasma-treated fiber samples (same preparation as for the reference) and then with precured fiber coating for min and belt-pressed with the speed: (**b**) 20 min and 1 m/min; (**c**) 30 min and 1 m/min; (**d**) 40 min and 2.5 m/min.

**Table 1 materials-17-04244-t001:** Thermal analysis of the ELO resin via DSC and TGO: Glass transition temperature (T_g_), reaction and decomposition enthalpy (ΔH_r_ and ΔH_d_), the temperature of the reaction and decomposition and their respective intervals (T_r_ and T_d_).

T_g_ [°C]	ΔH_r_ [J/g]	T_r_ [°C] and Reaction Interval	ΔH_d_ [J/g]	T_d_ (DSC) [°C] and Decomposition Interval	T_d_ (TGA) [°C]	Weight Loss [%]
96.1	162	155.5 (108–250)	91.8	361.6 (315–412)	358.3	98.9

## Data Availability

The original contributions presented in the study are included in the article, further inquiries can be directed to the corresponding author.

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
