# Peer review of "Natural Fiber-Reinforced Composite Incorporated with Anhydride-Cured Epoxidized Linseed-Oil Resin and Atmospheric Pressure Plasma-Treated Flax Fibers"

_materials, 2024, doi:10.3390/ma17174244_

Round 1
Reviewer 1 Report
Comments and Suggestions for Authors
This work is nice. My overall recommendation is minor revision. Below are my comments.
11. Could the authors comment more on what applications the developed NFRC technique can potentially work in? Please comment on both established and emerging applications.
2. Could the authors illustrate more on the physical significance of the performance metrics used in this work for better readership of general audience? E.g., elastic modulus of 18.3 GPa, and dispersive surface energy of 40 mN/m.
3. Could the authors quantify the repeatability or failure rate of the fabrication process shown in Fig. 1?
Author Response
Dear Ms./Mr. Reviewer,
Thank you very much for your review. I have provided the answers below.
Comment 1: Could the authors comment more on what applications the developed NFRC technique can potentially work in? Please comment on both established and emerging applications.
Response 1:
Page 2, lines 45-50.
Their lighter weight brings advantages like reduced fuel consumption and improved recycling opportunities. NFRC are commonly used in the automotive industry, particularly in door panels and linings, often combined with materials like epoxy, thermoset resin, PLA, and polypropylene (PP). Beyond the automotive sector, fiber composites have the potential for broad applications, ranging from aircraft components and electronics to construction.
Comment 2: Could the authors illustrate more on the physical significance of the performance metrics used in this work for better readership of general audience? E.g., elastic modulus of 18.3 GPa, and dispersive surface energy of 40 mN/m.
Response 2:
Page 5, lines 184-187.
The main motivation for the plasma pre-treatment was to achieve surface cleaning (i.e., to remove organic contaminations such as waxes or fats) and surface functionalization to improve interface strength, increase the surface area, and activate all bonding mechanisms responsible for adhesion - mechanical, physical, and chemical).
Page 9, lines 270-274
The tensile properties of NFRC are mostly affected by the interfacial adhesion between the resin and fibers, fiber volume fraction in the matrix resin. The value of the last one could be determined by 40 – 50 wt% of fiber content for injection molded thermoplastic matrix and leads to the load distribution to fibers and not matrix and therefore decreased risk of failure of the whole composite.
Comment 3: Could the authors quantify the repeatability or failure rate of the fabrication process shown in Fig. 1?
Response 3:
For the mechanical testing composite samples were prepared, varying precuring time (0 (not coated), 20, 30 and 40 min) at processing speed (1 and 2 m/min). Overall whole process was repeated 18 times (3 samples*6 variation except of reference (also 3 samples)).
Page 4, lines 177-178.
For each set of conditions (curing times and processing speed of the press) three samples were prepared.

Reviewer 2 Report
Comments and Suggestions for Authors
In my opinion, the article: "Natural fiber-reinforced composite incorporated with anhydride-cured epoxidized linseed oil resin and atmospheric pressure plasma treated flax fibers", is quite well structured from an experimental point of view. On this regards, the proposed procedure to highlight the effect of the performed treatments on the composites properties can be agreed, as well as the experimental testing activity performed to identify the processing window. Moreover, it has been accurately described and understandable.
On the other hand, important information and data, needed to understand the accuracy of the performed measures and the results reliability, are missing, despite the widest part of the results themselves can be agreed. In my opinion they can be obtained also from the already performed experimental part. English language is good.
For this reason the manuscript deserves to be published after a "minor" revision.
Here some (mandatory) question to be solved:
· Pag 3 - raws 105 - 107. The authors claim: "Raw flax tape (density 110 g/m3) was supplied by Ecotechnilin and non-woven flax (density 270 g/m3) by Ecco Gleitwing".
- The authors are recommended to describe the fibers orientation in the purchased tape;
· Pag 4 - raws 160 - 162. The authors claim: "The tensile strength of the prepared composite was DIN EN ISO 527 at a rate of 2 mm/min and an initial length of 50 mm".
- Probably there are missing words, hence they have to rewrite the sentence (i.e. "The tensile strength of the prepared composite was investigated using the DIN EN ISO 527 as a standard, at a rate of 2 mm/min and an initial length of 50 mm").
- The authors claim that the initial length is 50 mm. What do they mean with the term: "initial length"?. Is it the initial gage length (L0)?
- The authors have also to specify the used testing machine (together with the features of the load cell) and to clarify if they use external or internal strain gages;
- The authors are recommended to better describe the used specimens, as geometry and measures;
- The author correctly talk about the Tensile strength, as this characteristic is directly affected by the fiber/matrix interface, which allows the matrix to effectively share the applied loads between the fibers. Tensile strength and the corresponding deformation are more directly affected by the fiber/matrix interface then the Young Modulus, but the authors reported only the measured values of the latter characteristic.
For this reason, the authors are recommended to introduce into the manuscript also references to the developed composites tensile strength, at least and to discuss the obtained results also in relation to this property.
· Pag 7 - raws 229 - 221. The authors claim: "As can be seen from Figure 6, the composite containing the ELO coating cured for 20 min that was further belt-pressed with PLA at speeds of 2.5 m/min showed the highest elastic modulus value, i.e., 18.3 GPa".
- The mentioned values of the Young Modulus are very high for a natural (Flax) fiber composite. They are compatible with laminates based on unidirectional fibers, whose volume fraction is about 60%. The latter is a very critical target for the investigated fibers, even in the case of thermosetting matrices based unidirectional (flax) fibers, processed by mean of an autoclave.
In the case of composites based on non woven mat, the obtained results appear really too high.
For this reason, the authors are recommended to provide information on the fiber volume fraction for the whole range of the produced laminates and the value of the young modulus for the used fibers.

Author Response
Dear Ms./Mr. Reviewer,
Thank you very much for your review. I have provided the answers below.
Comment 1: Pag 3 - raws 105 - 107. The authors claim: "Raw flax tape (density 110 g/m3) was supplied by Ecotechnilin and non-woven flax (density 270 g/m3) by Ecco Gleitwing".
- The authors are recommended to describe the fibers orientation in the purchased tape;
Response 1: Raw flax tape (density 110 g/m3, unidirectional) was supplied by Ecotechnilin and non-woven flax (density 270 g/m3, cross-stitched) by Ecco Gleitwing.
Comment 2: Pag 4 - raws 160 - 162. The authors claim: "The tensile strength of the prepared composite was DIN EN ISO 527 at a rate of 2 mm/min and an initial length of 50 mm".
- Probably there are missing words, hence they have to rewrite the sentence (i.e. "The tensile strength of the prepared composite was investigated using the DIN EN ISO 527 as a standard, at a rate of 2 mm/min and an initial length of 50 mm").
- The authors claim that the initial length is 50 mm. What do they mean with the term: "initial length"?. Is it the initial gage length (L0)?
- The authors have also to specify the used testing machine (together with the features of the load cell) and to clarify if they use external or internal strain gages;
- The authors are recommended to better describe the used specimens, as geometry and measures;
Response 2: Static tensile tests were conducted at 23 °C using a 2.5 kN load cell on Zwick Z050 - ZwickRoell. Following the DIN EN ISO 527 standard, the tensile strength of the prepared composites was assessed at a testing speed of 2 mm/min. Each specimen measured 150 mm in length and 15 mm in width, with an external clamp gauge (makroXtens – ZwickRoell) length of 50 mm and a length of 110 mm between the grips during testing. For each set of conditions (curing times and processing speed of the press) three samples were prepared. The morphology of the cryo fractured surfaces of the materials were examined by scanning electron microscopy (SEM) type Quanta 3D FEG (FEI) with an Everhart-Thornley-Detector and acceleration voltage of 5 kV.
Comment 3: The author correctly talk about the Tensile strength, as this characteristic is directly affected by the fiber/matrix interface, which allows the matrix to effectively share the applied loads between the fibers. Tensile strength and the corresponding deformation are more directly affected by the fiber/matrix interface then the Young Modulus, but the authors reported only the measured values of the latter characteristic.
For this reason, the authors are recommended to introduce into the manuscript also references to the developed composites tensile strength, at least and to discuss the obtained results also in relation to this property.
Response 3: Page 8, lines 253-265
For better understanding the mechanical performance of the composite, the tensile test results were compared with values reported in the literature. The Young’s modulus of prepared ELO composite was found to be higher than the 7.3 GPa observed for PLA/40 wt. % flax composites. The same authors studied the effect of adding a triacetin plasticizer (Tri) to PLA, which did not enhance mechanical properties (7.3 GPa for PLA/5 wt. % Tri/40 wt. % flax). Both of these composites were prepared using extrusion, a process that can subject fibers to shearing, potentially leading to fiber degradation and lower mechanical performance. In the current study, the fibers were subjected to multiple heating cycles during ELO curing and belt pressing, yet the results remained relatively high and should be carefully considered in each specific case. The high standard deviation (S.D.) of tensile stress could be explained by not uniform loading during the composite preparation and could be improved, varying manufacturing techniques, compression modeling, hand lay-up ore belt-press.
Comment 4: Pag 7 - raws 229 - 221. The authors claim: "As can be seen from Figure 6, the composite containing the ELO coating cured for 20 min that was further belt-pressed with PLA at speeds of 2.5 m/min showed the highest elastic modulus value, i.e., 18.3 GPa".
- The mentioned values of the Young Modulus are very high for a natural (Flax) fiber composite. They are compatible with laminates based on unidirectional fibers, whose volume fraction is about 60%. The latter is a very critical target for the investigated fibers, even in the case of thermosetting matrices based unidirectional (flax) fibers, processed by mean of an autoclave.
In the case of composites based on non woven mat, the obtained results appear really too high.
For this reason, the authors are recommended to provide information on the fiber volume fraction for the whole range of the produced laminates and the value of the young modulus for the used fibers.
Response 4: Page 4, line 153
The fiber content was held constant, 40 wt. %.
Also Page 8, lines 243-265

Reviewer 3 Report
Comments and Suggestions for Authors
The manuscript titled “Natural fiber-reinforced composite incorporated with anhydride-cured epoxidized linseed oil resin and atmospheric pressure plasma treated flax fiber” by Savicheva, S.; et al. is a scientific work where the authors assessed the positive impact of plasma treatment on the surface free energy and chemistry modification of flax fibers improving the wetting with PLA matrices. For it, many complementary techniques were devoted in this research. This is a topic of growing interest. However, it exists some points that need to be addressed (please, see them below detailed point-by-point) to improve the scientific quality of the submitted manuscript paper before this article will be consider for its publication in Materials.
1) The authors should consider to add the term “flax” in the keyword list.
2) “Composite materials are a highly versatile type (…) stiffness and strength” (lines 35-40). Could the authors provide quantitative insights about the worldwide economic impact of composite manufacturing sector? This will significantly aid the potential readers to better understand the significance of this work.
3) “A crucial (…) NFRC (…) matrix” (lines 44-46). Even if it was already defined during the abstract section, the full-term should be stated. Then, the abbreviation “NFRC” should be placed between brackets. This comment should be taken into account for the rest of the main manuscript body text.
4) “Natural fibers are composed of cellulose and hemicellulose (…) A composite based on natural fibers (…) moisture uptake (…) microcrakcs in the matrix and further composite degradation (…) further composite degradation” (lines 46-52). Here, even if I agree with the information provided in these statements by the authors, it should be remarkable to mention some examples about the existing interfacial forces exerted between flax fibers and PLA [1] and PBS [2] matrices. Furthermore, flax fibers exhibit larger interactions with the respective matrices compared to other lignified fibers as kenaf or jute.
[1] Charca, S.; Jiao-Wang, L.; Santiuste, C. Influence of Reinforcement Architecture on Behavior of Flax/PLA Green Composites under Low-Velocity Impact. Materials 2024, 17, 2958. https://doi.org/10.330/ma17122958.
[2] Marcuello, C.; Chabbert, B.; Berzin, F.; Bercu, N.B.; Molinari, M.; Aguié-Béghin, V. Influence of Surface Chemistry of Fiber and Lignocellulosic Materials on Adhesion Properties with Polybutylene Succinate at Nanoscale. Materials 2023, 16, 2440. https://doi.org/10.3390/ma16062440.
5) “2.5.5. Characterization” (lines 142-162). Some information should be furnished about the acquisition of scanning electron microscopy (SEM) images. Same comment for the tensile tests conducted to determine the Young’s modulus. Furthermore, some physical equations should be detailed to explain how to extract these physical properties.
6) Figure 2 (line 177). Some statistical analysis should be devoted in order to discern if the observed differences among the tested conditions are significally different. Same comment for the Figure 5 (line 221) and Figure 6 (line 232).
7) “3.3. Mechanical characterization of the prepared NFRC” (line 223-244). Here, the authors should show the tensile stress-strain curves for all the examined samples
8) Conclusions (lines 245-265). This section perfectly remarks the most relevant outcomes found by the authors in this work and the promising future perspectives. It should be desirable to add a brief statement to discuss about the potential future action lines to pursue the topic covered in this research.
Comments on the Quality of English LanguageThe manuscript is generally well-written albeit it may be desirable to check it in order to polish those final details susceptible to be improved.
Author Response
Dear Ms./Mr. Reviewer,
Thank you very much for your review. I have provided the answers below.
Comment 1: The authors should consider to add the term “flax” in the keyword list.
Response 1: Corrected
Comment 2: “Composite materials are a highly versatile type (…) stiffness and strength” (lines 35-40). Could the authors provide quantitative insights about the worldwide economic impact of composite manufacturing sector? This will significantly aid the potential readers to better understand the significance of this work.
Response 2: Page 2, lines 51-54
The forecast for the consumption of natural fiber composites worldwide, prepared by Beijing Yubo Zhiye Market Consulting Co. showed three significant regions for the development of natural fiber composites, North America, China and Europe, from approximately 200 ten thousand ton in 2021 up to near 300 in 2026.
Comment 3: “A crucial (…) NFRC (…) matrix” (lines 44-46). Even if it was already defined during the abstract section, the full-term should be stated. Then, the abbreviation “NFRC” should be placed between brackets. This comment should be taken into account for the rest of the main manuscript body text.
Response 3: Corrected
Comment 4: “Natural fibers are composed of cellulose and hemicellulose (…) A composite based on natural fibers (…) moisture uptake (…) microcrakcs in the matrix and further composite degradation (…) further composite degradation” (lines 46-52). Here, even if I agree with the information provided in these statements by the authors, it should be remarkable to mention some examples about the existing interfacial forces exerted between flax fibers and PLA [1] and PBS [2] matrices. Furthermore, flax fibers exhibit larger interactions with the respective matrices compared to other lignified fibers as kenaf or jute.
[1] Charca, S.; Jiao-Wang, L.; Santiuste, C. Influence of Reinforcement Architecture on Behavior of Flax/PLA Green Composites under Low-Velocity Impact. Materials 2024, 17, 2958. https://doi.org/10.330/ma17122958 .
[2] Marcuello, C.; Chabbert, B.; Berzin, F.; Bercu, N.B.; Molinari, M.; Aguié-Béghin, V. Influence of Surface Chemistry of Fiber and Lignocellulosic Materials on Adhesion Properties with Polybutylene Succinate at Nanoscale. Materials 2023, 16, 2440. https://doi.org/10.3390/ma16062440 .
Response 4: Thank you for sharing with this researches. I agree, it should be mentioned.
Page 2, lines 63-64
Other factor which could influence on the final composite performance is reinforcement architecture of fibers themself [] or their lignin content [].
Comment 5: “2.5.5. Characterization” (lines 142-162). Some information should be furnished about the acquisition of scanning electron microscopy (SEM) images. Same comment for the tensile tests conducted to determine the Young’s modulus. Furthermore, some physical equations should be detailed to explain how to extract these physical properties.
Response 5: Thank you very much for your comment. Information about SEM and Tensile measurements are added. Page 4-5, lines 173-182
The aim of the current study was to demonstrate the improved compatibility between the reinforcement fibers and the polymer matrix by modifying the fibers with plasma treatment and applying an intermediate layer. This research encompasses various fields, including the investigation of fiber surface chemistry, the preparation of a thermoset layer using epoxidized linseed oil, and the mechanical evaluation of the resulting composite. During the preparation of paper, one of the goals was to present a comprehensive and clear overview of the relevant scientific aspects—chemistry, material science, and mechanics. Sufficient references were provided throughout to ensure that readers could grasp the main concepts, gain valuable insights relevant to their field, and, if needed, explore the topic further through additional sources. The information is intended to be thorough and informative without being overwhelming, that could happen with adding additional physical calculations.
Comment 6: Figure 2 (line 177). Some statistical analysis should be devoted in order to discern if the observed differences among the tested conditions are significally different. Same comment for the Figure 5 (line 221) and Figure 6 (line 232).
Response 6: The mean and standard deviation of Surface energy values and mechanical properties of the composite are presented in Table 2.
Comment 7: “3.3. Mechanical characterization of the prepared NFRC” (line 223-244). Here, the authors should show the tensile stress-strain curves for all the examined samples
Response 7: The mechanical properties of prepared NFRC, such as Young’s modulus, tensile stress and elongation to break are presented in the Table 2.
Comment 8: Conclusions (lines 245-265). This section perfectly remarks the most relevant outcomes found by the authors in this work and the promising future perspectives. It should be desirable to add a brief statement to discuss about the potential future action lines to pursue the topic covered in this research.
Response 8: Page 10, lines 306-311.
Future challenges in the development of NFRC could focus on enhancing thermoset preparation, such as reducing the amount of anhydride and catalyst, and improving the reactivity of the main thermoset component, epoxidized linseed oil by replacing it with more chemically reactive acrylated epoxidized plant oil, which contains not only epoxy rings but also hydroxyl groups. Additionally, conducting fracture tests on anti-parallel oriented laminates could provide deeper insights into their mechanical performance.

Round 2
Reviewer 3 Report
Comments and Suggestions for Authors
The authors did a great deal of effort to cover all the suggestions raised by the authors. For this reason, the scientific manuscript quality was greatly improved. Based on the significance of the most relevant outcomes found in this research, I warmly endorse this work for further publication in Materials journal.